# Ultrasonic Diagnosis of Intestinal Obstruction in Neonates-Original Article

**DOI:** 10.3390/diagnostics13050995

**Published:** 2023-03-06

**Authors:** Binbin Guo, Lin Pang, Chang Liu, Xiaoya Chen, Qiang Qiao, Cong Zhang

**Affiliations:** 1Department of Ultrasound, The Sixth Affiliated Hospital of Harbin Medical University, Harbin 150076, China; 2Department of Ultrasound, Harbin Finance University, Harbin 150030, China; 3Department of Ultrasound, The First Affiliated Hospital of Harbin Medical University, Harbin 150007, China

**Keywords:** ultrasound, intestinal obstruction, neonates, intestinal malrotation, intestinal atresia

## Abstract

Background: Intestinal obstruction in neonates is a common problem that requires prompt diagnosis and treatment, and ultrasound could be a potential tool for it. The purpose of this study was to investigate the accuracy of ultrasonography in diagnosing and identifying the cause of intestinal obstruction in neonates, the corresponding ultrasonic manifestations, as well as to utilize the diagnostic method. Methods: We conducted a retrospective study of all neonatal intestinal obstruction in our institute between 2009 and 2022. The accuracy of ultrasonography in the diagnosis of intestinal obstruction and the identification of its etiology was compared with the results of operation as the gold standard. Results: The accuracy of the ultrasonic diagnosis of intestinal obstruction was 91%, and the accuracy of the ultrasonic etiological diagnosis of intestinal obstruction was 84%. The main ultrasound findings for the neonatal intestinal obstruction were dilation and high tension of the proximal bowel and collapse of the distal intestinal. Other major manifestations were the presence of corresponding diseases causing intestinal obstruction at the junction of the dilated and collapsed bowel. Conclusions: Ultrasound has the advantages of being a flexible multi-section dynamic evaluation and a valuable tool to diagnose and identify the cause of intestinal obstruction in neonates.

## 1. Introduction

When the newborn has vomiting and abdominal distension, it is necessary to be highly alert to diseases that may endanger life and have serious complications. Neonatal intestinal obstruction is one of them. The condition can be significantly aggravated in a short time. Once complications of neonatal intestinal obstruction occur, the mortality rate is as high as 15–70% [1,2]. Therefore, neonatal intestinal obstruction needs early surgical treatment, and early diagnosis can save the lives of children and reduce medical disputes. Compared with traditional X-ray examinations, ultrasound diagnosis has the advantages of dynamic monitoring, being fast and timely, and producing no radiation. The purpose of this study is to explore the accuracy of ultrasound in diagnosing and identifying the causes of neonatal intestinal obstruction, the corresponding ultrasound manifestations, and diagnostic method so that more doctors can use ultrasound to accurately diagnose neonatal intestinal obstruction and its causes.

At present, there are few studies on the application of ultrasound to diagnose intestinal obstruction and determine the cause of intestinal obstruction. Most of them only discussed one kind of disease that induced intestinal obstruction, such as congenital hypertrophic pyloric stenosis, intestinal malrotation, or midgut volvulus. However, relatively little is known about comparing the ultrasonic manifestations of different causes of intestinal obstruction, and there are few studies summarizing and giving diagnostic ideas. This study also aims to fill up the gap and provides a comprehensive diagnosis of common causes of neonatal intestinal obstruction for clinicians and ultrasound doctors.

## 2. Materials and Methods

We conducted a retrospective study of all neonates with a diagnosis of intestinal obstruction in our institute between 2009 and 2022. Patients without surgically confirmed or preoperative ultrasound were excluded from the study. The neonates’ sex, gestational age, clinical manifestations, findings and diagnosis of ultrasound, and operation results were all collected.

Each neonate was scanned from the lower esophagus to the pylorus, and then the duodenal bulb, descending; horizontal and ascending parts were scanned continuously according to the anatomical position; and the left upper abdominal jejunum area was scanned to the left lower abdomen; then, scan the small intestine without omission in the middle abdomen and the right abdomen, search for the ileocecal region and ascending colon on the right abdomen, search for the descending colon on the outside of the left abdomen, continue to the sigmoid colon, look for the rectum behind the bladder, and observe whether there is an abdominal cavity in the diaphragm contents, check thickness of pyloric muscle, the course of duodenum, and observe the bowel diameter, tension, and content, etc. Then, transect along the abdominal aorta in the mid abdomen and observe the relationship between the superior mesenteric artery and the proximal end of the superior mesenteric vein. Serial images were obtained using MyLab30 (Esaote, Genoa, Italy) with a 3–11 MHz linear transducer. All examinations and diagnoses were performed by two sonographers who were trained and experienced in pediatric ultrasound.

Ultrasonic diagnosis ideas and methods are shown in Figure 1.

The accuracy of ultrasound in the diagnosis of intestinal obstructions and the identification of its etiology were compared with the results of operation as the gold standard.

All data were collected, tabulated, and calculated, and numerical data were presented as mean ± SD.

## 3. Results

86 cases were included with a total of 91 neonates. Another five cases were excluded, including two neonates without being confirmed by surgery and three neonates due to lack of a preoperative ultrasound. Table 1 summarized the distribution of patients with different diseases leading to neonatal intestinal obstruction and the number of ultrasonic diagnoses in this study.

### 3.1. The Accuracy of Ultrasonography in Intestinal Obstruction of Neonates

#### 3.1.1. The accuracy of Ultrasonography in Diagnosing of Intestinal Obstruction in Neonates

In general, with surgical outcomes as the gold standard, 79 cases were accurately diagnosed via ultrasound among 86 cases of intestinal obstruction, and the accuracy rate of ultrasonic diagnosis of an intestinal obstruction was 91%. Intestine atresia was the most common cause of neonatal ileus. Jejunal atresia was the most common atresia followed by ileal atresia. There were seven cases of intestinal obstruction without definite and accurate diagnosis via ultrasound, including one case of anal atresia, one case of duodenal stricture, two cases of circular pancreas, and three cases of congenita megacolon (Table 1).

#### 3.1.2. The Accuracy of Ultrasonography in Identifying the Cause of Intestinal Obstruction in Neonates

72 cases were exactly identified as causes via ultrasound among 86 cases of intestinal obstruction with surgical outcomes as the gold standard, and the accuracy rate of ultrasonic etiological diagnosis of the intestinal obstruction was 84%. In detail, the diagnostic accuracy of ultrasound is different in various diseases causing neonatal intestinal obstruction (Table 1). Seven cases of intestinal obstruction without definite and accurate diagnosis of causes via ultrasound, including one case of ileal atresia, one case of anal atresia, three cases of duodenal stricture, and two cases of a circular pancreas.

### 3.2. The Ultrasonic Manifestations

#### 3.2.1. The Ultrasonic Signs of Intestinal Obstruction

The main ultrasound findings for neonatal intestinal obstructions were dilation and high tension of the proximal bowel and collapse of the distal intestinal in 79 patients (91%). Proximal bowel width was 1.9 ± 0.6 cm and the distal bowel width was 1.1 ± 0.5 cm. The second major manifestation was the presence of corresponding diseases causing ileus at the junction of the dilated and collapsed bowel.

#### 3.2.2. The Corresponding Ultrasound Findings of Diseases Leading to Neonatal Ileus

1. Intestinal atresia: The main ultrasound findings for neonatal intestinal obstructions were also dilation of the proximal bowel and collapse of the distal intestine. It is characterized by an extremely small, fetal colon (0.7 ± 0.2 cm) in cases of jejunal atresia, ileal atresia, and duodenal atresia (Figure 2). It showed dilatation of the colon and rectum after 24 h of life in cases of anal atresia.

2. Duodenal obstruction: The primary ultrasound signs of a congenital duodenal obstruction were liquid contents retained in the stomach and the proximal lumen of the duodenum and collapse of the distal intestinal obstruction. The typical ultrasound sign of an annular pancreas was the pancreas surrounding the second part of the duodenum. Intestinal malrotation with midgut volvulus had a specific sonographic feature called a “swirl sign” (Figure 3 and Figure 4).

3. Diaphragmatic hernia: the presence of the intestinal canal inside the thorax (Four out of four, three left and one right) was the main ultrasonographic manifestation of it. Two patients also observed spleen inside the thorax (two of out four).

4. Congenital hypertrophy pyloric stricture: the major ultrasonographic sign of this is the pyloric muscle layer thickened. The length of pyloric muscle was 20.1 mm, the diameter was 12.5 mm, and the muscle thickness was 4.8 mm (Figure 5).

## 4. Discussion

Intestinal obstruction of the digestive tract is one of the most common causes of surgery on neonates. Results of an early study showed that intestinal perforation with meconium peritonitis in the neonatal ileus, which carries a high mortality rate, is also common [1,2,3]. Early surgical treatment usually achieves good therapeutic effect and prognosis. Accordingly, early and accurate diagnosis of intestinal obstruction is important to improve prognosis and even save lives. Ultrasonography, which is radiation-free, is more suitable for children, especially newborns. A fairly accurate diagnosis of gastrointestinal obstruction is the recommended alternative.

X-ray is a traditional and essential method for the diagnosis of ileus, especially in adults. In addition to conventional standing plain abdominal radiography, upper and lower gastrointestinal radiography are also very useful for the location of ileus and the determination of its cause. Ultrasonography, which is radiation-free, is more suitable for children, especially newborns. The abdominal wall of newborns is so thin that the sonographer can clearly observe the internal structure. After a systematic and in-depth study, an experienced sonographer can not only diagnose intestinal obstruction but also identify most of its causes. In our study, although the children also underwent X-ray examinations, we took the surgical results as the gold standard and only collected ultrasonic data to study the accuracy of the ultrasonic diagnosis of neonatal ileus. The results show that ultrasound is a reliable alternative method for diagnosing gastrointestinal obstruction in neonates.

With surgical outcomes as the gold standard, 79 cases were accurately diagnosed via ultrasound among 86 cases of intestinal obstruction; the accuracy rate for the ultrasonic diagnosis of an intestinal obstruction was 91% in our study. Among the 86 cases of neonatal intestinal obstruction, there were fifteen cases of jejunal atresia, fourteen cases of ileal atresia, twelve cases of duodenal atresia, eight cases of anal atresia, twelve cases of intestinal malrotation with midgut volvulus, ten cases of a duodenal stricture, five cases of a circular pancreas, one case of intestinal atresia with intestinal malrotation, one case of intestinal atresia with midgut volvulus, four cases of diaphragmatic hernia, one case of a congenital hypertrophy pyloric stricture, and three cases of a congenital megacolon.

Atresia is noted to be the commonest cause of intestinal obstruction in neonates worldwide [4]. Atresia is so common in ileus that the accuracy rate of an ultrasonic diagnosis of the intestinal obstruction is high. Seven cases of intestinal obstruction without definite and accurate diagnosis were also mainly due to the absence of the typical manifestations described above, including one case of anal atresia, one case of a duodenal stricture, two cases of a circular pancreas, and three cases of a congenital megacolon. Because the ultrasound examination of the newborn was conducted after gastrointestinal decompression and lower digestive tract angiography, the dilatation of the stomach and duodenum and the shrinkage of the colon and rectum were not obvious in cases of duodenal obstruction. Due to a lack of experience, one case of anal atresia was missed in our study because it was examined within 24 h of birth. It has been reported that imaging to assess the position of distal rectal gas to determine the type of anorectal malformation should be performed after a period of time (24 h after birth) to allow meconium to metastasize to the distal end [5,6,7].

In addition to the intestinal atresia mentioned above, the causes of neonatal intestinal obstruction include a congenital hypertrophic pyloric stricture, duodenal stricture, annular pancreas, congenital intestinal malrotation, meconium intestinal obstruction, megacolon, congenital anal atresia, etc. Among the various causes, small bowel atresia is one of the most common etiologies of intestinal obstruction in our study, which is in agreement with the reported literature [8,9,10]. As discussed above, an ultrasound can diagnose intestinal atresia based on the typical manifestations of the proximal bowel dilation and distal bowel collapse. Intestinal atresia can occur in any location on the small bowel as a solitary lesion or even multiple lesions. The incidence of duodenal atresia has been estimated as 1 in 6000–10,000 live births. Jejunoileal atresia is seen in 1 in 5000 to 1 in 14,000 live births [11,12]. In our study, jejunum atresia was the most common atresia in our study, followed by ileal atresia and duodenal atresia.

Congenital duodenal obstruction is one of the most common anomalies in newborns, affecting approximately 1 in 2500 to 10,000 live births, and accounting for nearly half of all cases of neonatal intestinal obstruction [13]. In addition to duodenal atresia discussed above, other causes include a duodenal diaphragm, annular pancreas, congenital bands, and abnormalities of intestinal rotation, such as midgut volvulus, duplication cysts, and a preduodenal portal vein. In our study, among the eighty-six cases of neonatal intestinal obstruction, there were twelve cases of duodenal atresia, ten cases of a duodenal stricture, five cases of an annular pancreas, and twelve cases of intestinal malrotation with midgut volvulus. 

The typical ultrasound signs of a duodenal obstruction are liquid contents retained in the stomach and proximal lumen of the duodenum and the collapse of the distal intestinal obstruction. Other major ultrasonic manifestations were the presence of corresponding diseases causing ileus in the duodenum. Some of them are not easily observed. The diaphragm and pancreas surrounding the duodenum and congenital bands that causes duodenal stricture are not easily detected. The reasons for missed diagnosis are as follows: the duodenal septum is not always visible, and membrane-like hypoecho between the dilated intestines may be observed after fluid has filled through the stenosis to the distal end of the stenosis. So, the sonographer should observe the dilated bowel for a long time after the newborn has had water or milk. A circular pancreas usually has only a thin layer of glands surrounding the duodenum which makes it difficult to see the pancreatic tissue surrounding the duodenum, resulting in it being missed by the ultrasonographer.

Some ultrasonic signs are specific for diagnosis. Intestinal malrotation with midgut volvulus has specific sonographic features, such as the ‘whirlpool’ sign, as not only the intestinal tract but also the mesenteric vessels are volvulated in midgut volvulus. The ‘whirlpool’ sign, referring to the appearance of the SMV wrapping in a clockwise manner around the SMA, has been used as evidence for malrotation with midgut volvulus [14,15]. Based on the typical sign, ultrasonography is regarded as a highly accurate examination for the diagnosis of midgut volvulus, and the diagnosis of midgut volvulus with positive and negative predictive values with US was 100% [16,17]. Consistent with the findings above, the whirlpool sign was observed in all cases of midgut volvulus in our study. Because the volvulated intestinal tract is at risk of extensive ischemic necrosis, which can be fatal, an accurate diagnosis must be made promptly, and the patient must undergo surgery without delay. We agree with the proposition that further imaging investigations are often not needed with the classic sonographic appearance of the whirlpool sign; in addition, the surgeon should be alerted to plan surgery which is in agreement with the reported literature [18,19,20]. However, although midgut malrotation has the typical ultrasonic sign, it is only a common complication of malrotation. Intestinal malrotation without midgut malrotation is often missed.

Congenital causes of intestinal obstruction can occur either in isolation or in association with other congenital anomalies. There is intestinal atresia with malrotation and intestinal atresia with intestinal volvulus in our study. So, the sonographer should also pay attention to the presence of other associated malformations after discovering one intestinal malformation during the examination.

Diaphragmatic hernias causing ileus are usually so large that the bowel in addition to the parenchymal organs can herniate into them. This type of diaphragmatic hernia has specific signs indicating the presence of the intestinal canal inside the thorax. A diaphragmatic hernia occurs in approximately 1 in 2500 to 5000 infants [20]. Researchers found that an ultrasound was useful for detecting the location and defect size of the diaphragmatic hernia and determining optimal surgical management [21,22,23,24]. We compared the accuracy of the US diagnosis with that of the surgical diagnosis for the location of the diaphragmatic hernia and the hernial contents. The results of our research show that the location and contents of the diaphragmatic hernia were both consistent between the US and surgical findings. The most common ultrasonographic manifestation of the diaphragmatic hernia was the presence of intestinal tubes in the thoracic cavity in our study. Nevertheless, we should be aware that the accuracy of an ultrasound in diagnosing a diaphragmatic hernia without causing intestinal obstruction will not be so high.

A congenital megacolon is the main genetic cause of a functional intestinal obstruction. Congenital megacolon (Hirschsprung’s disease) can be treated surgically in the neonatal period, infancy, childhood, or even adulthood [25,26,27,28]. They are usually advised to be treated symptomatically and rarely surgically during the neonatal period. In our study, there were three cases of a megacolon being operated on in the neonatal period, all of which were missed via ultrasound.

Hypertrophic pyloric stricture (HPS) is the most frequent, surgically correctable lesion causing vomiting in infants. As a hypertrophic pyloric stricture is a developmental disorder that is frequently seen in male infants between 4 and 6 weeks old, it rarely presents in the neonatal period [29]. There was only one case in our study. The diagnostic criteria for HPS are a pyloric length more than 14.5 mm and muscle thickness more than 3 mm [30]. Consistent with previous studies, the length of the pyloric muscle was 20.1 mm and the muscle thickness was 4.1 mm in our study.

In order to facilitate the researchers to copy our method and use ultrasound to clearly diagnose a neonatal intestinal obstruction, we summarized the flow chart of the diagnosis of an intestinal obstruction (Figure 1). We will explain the specific use of the flow chart as follows.

Firstly, we should determine whether there is an intestinal obstruction, that is, to determine whether there is a continuous expansion of the bowel, with the distal bowel being deflated. In a small bowel obstruction, the distal bowel usually refers to the colon. Therefore, an ultrasound should first distinguish the small intestine from the colon. Normally, it can be distinguished from four aspects. The first is the anatomical position. Second, the diameter of the colon is wider than that of the small intestine. Third, the colon has a colon pocket. The fourth is the intestinal contents; either empty or few contents in the small intestine showing a low echo, and the colon a high echo in the gas stool. When a small intestinal obstruction occurs, the small intestine is filled with a liquid echo, high tension, the diameter ratio of the colon located in the outermost part of the abdominal cavity is wide, and the colon is empty or has little air stool. The diameter of the small intestine is wider than that of the colon, and the tension is high, which can be confirmed as an intestinal obstruction.

Our second step was looking for the obstruction site. The position of obstruction can be preliminarily evaluated in three steps: (1) upper abdomen: evaluate the stomach; (2) left mid abdomen: jejunum and descending colon were evaluated; (3) right lower abdomen: ileocecal junction was evaluated. The duodenum and upper jejunum are usually considered as high intestinal obstructions, and the lower jejunum or ileum as a low intestinal obstruction. In theory, it is possible to determine whether the level of the small intestinal obstruction is located in the jejunum or the ileum based on the fact that there are more jejunum mucosal folds than with the ileum. After approximately determining the location of the obstruction, you must trace the junction of the dilated and deflated bowel at the site with the highest intestinal tension to determine the obstruction site.

Thirdly, look carefully for the cause at the site of obstruction. In intestinal obstruction is not a disease but a common manifestation of a variety of abdominal diseases. Diagnosis of etiologies relies on the identification of the bowel, localization of the obstruction, and typical sonographic findings. Take duodenal obstruction as an example; duodenal atresia is the easiest to recognize, it results in complete intestinal obstruction, and the distal bowel, jejunum, ileum, and colon are empty and deflated, devoid of contents, with neither gas nor fluid, as discussed for the Fetal-type colon, is small and deflated less than 0.5 cm, if there is such an ultrasound manifestation, even if the child has a small amount of content in the rectum or the child has meconium (atresia is in the late embryo, the content is formed in the middle of the embryo or secreted in the late stage, etc.); all these factors can also diagnose duodenal atresia. When it is found that the duodenum is obviously dilated and there is a small amount of content in the distal intestine, and the child has corresponding clinical symptoms such as vomiting one day after birth, the ultrasound doctor can consider the diagnosis of duodenal stenosis. Long-term observation may reveal a membrane-like hypoechoic in the bowel when the liquid passes through the stenosis and fills to the distal end of the stenosis, which can be used to diagnose duodenal stenosis caused by the duodenal septum.

## 5. Conclusions

Our study demonstrates that an ultrasound has the advantages of flexible multi-section dynamic evaluation and can provide accurate guidance for the analysis of a neonatal intestinal obstruction, which then helps to accurately diagnose ileus and determine the cause of neonatal ileus. A diagnostic flow chart may help more doctors use ultrasounds to diagnose a neonatal intestinal obstruction and its causes.

In a word, findings in this study confirm that an ultrasound is a reliable modality of choice in diagnosing a neonatal intestinal obstruction, in addition to identifying the etiology of the intestinal obstruction.

## Figures and Tables

**Figure 1 diagnostics-13-00995-f001:**
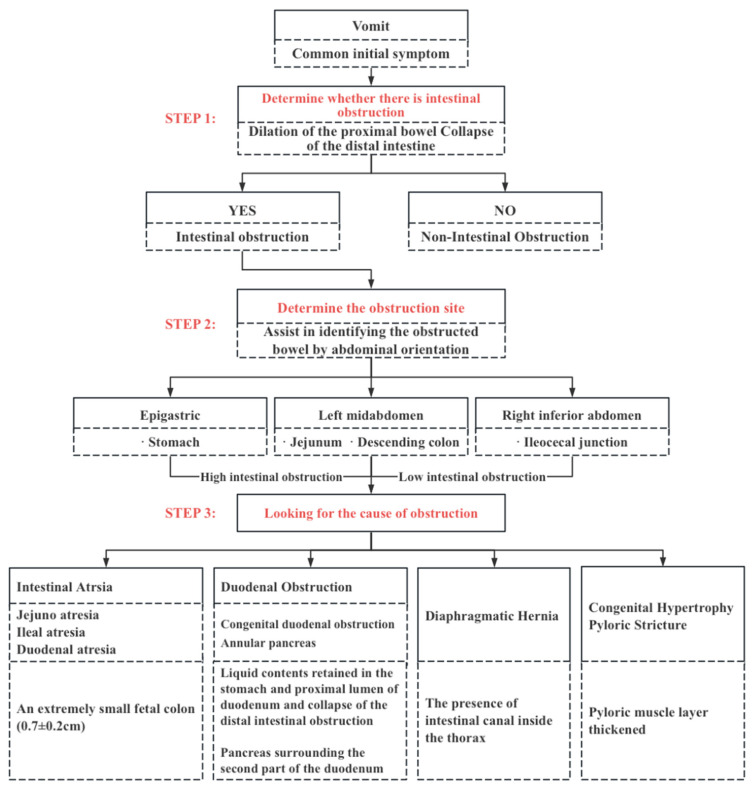
Flow chart of the diagnosis of intestinal obstructions.

**Figure 2 diagnostics-13-00995-f002:**
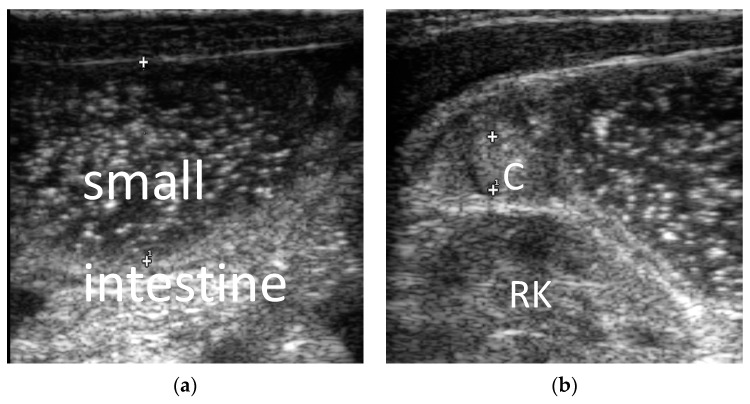
Male, 1 day. Ileum atresia. (**a**) Small intestine was dilated, the distance between the “+” and”+^1^” is the small intestine diameter; (**b**) ascending colon was shriveled and empty, showing a fetal colon, the distance between the “+” and”+^1^” is the colon diameter. C = colon. RK = right kidney.

**Figure 3 diagnostics-13-00995-f003:**
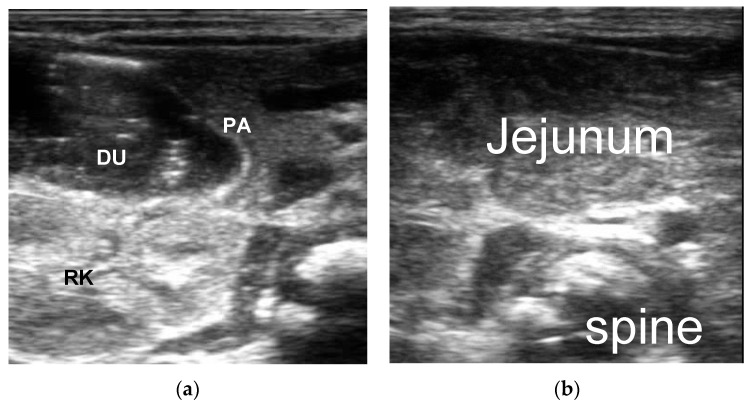
Male, 1 day. Duodenal atresia. (**a**) Descending part of duodenum was dilated, PA = Pancreas, DU = Duodenum, RK = Right Kidney; (**b**) complete atrophy of the left upper abdomen jejunum (no gas or contents); (**c**) Complete atrophy of the right abdomen ileum (no gas or contents).

**Figure 4 diagnostics-13-00995-f004:**
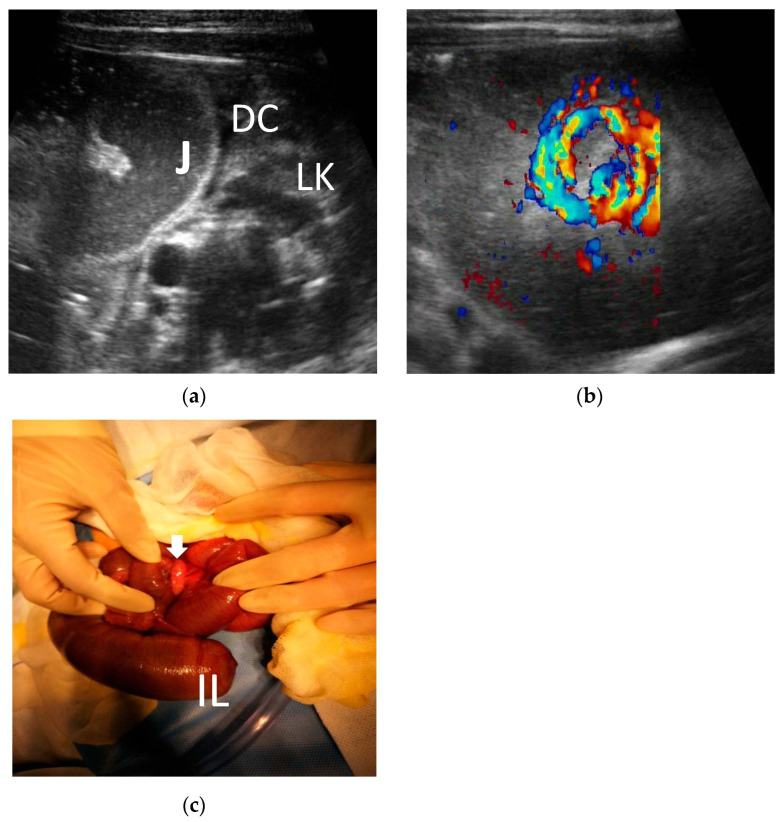
Male, 2 days. The distal ileum atresia with volvulus. (**a**) The jejunum was dilated, and the descending colon was empty showing a fetal colon. J = jejunum. DC = descending colon. LK = left kidney. (**b**) Blood swirl sign: superior mesenteric vein (SMV) wrapping in a clockwise manner around the superior mesenteric arteria (SMA). (**c**) One volvulus (white arrow) and distal ileum atresia were observed during the operation. IL = ileum.

**Figure 5 diagnostics-13-00995-f005:**
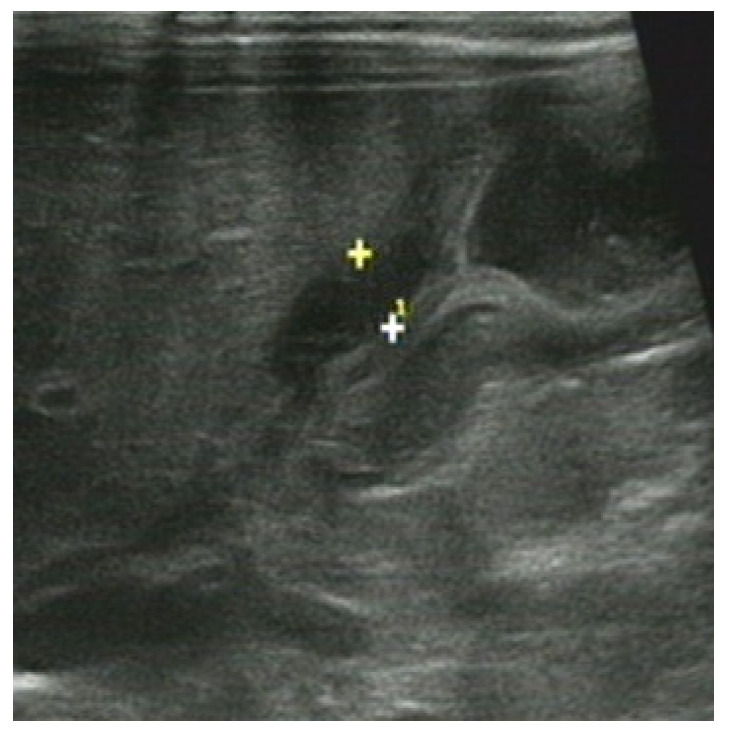
Male, 22 days. Congenital hypertrophy pyloric stricture: pyloric muscle layer thickened, The distance between "+" and "+^1^" is the thickness of the pyloric muscle layer.

**Table 1 diagnostics-13-00995-t001:** Patients’ distribution in different diseases leading to neonatal ileus.

Types of Malformations	N1	N2	N3
Jejunal atresia	15	15	15
Ileal atresia	14	14	13
Duodenal atresia	12	12	12
Anal atresia	8	7	6
Intestinal malrotation with midgut volvulus	12	12	12
Duodenal stricture	10	9	6
Annular pancreas	5	3	1
Intestinal atresia with malrotation	1	1	1
Intestinal atresia with intestinal volvulus	1	1	1
Diaphragmatic hernia	4	4	4
Congenital hypertrophy pyloric stricture	1	1	1
Congenital megacolon	3	0	0

N1 = Number of intestinal obstructions confirmed via surgery. N2 = Number of intestinal obstructions diagnosed via ultrasound. N3 = Number of causes of intestinal obstruction diagnosed via ultrasound.

## Data Availability

The data is available from the corresponding author upon reasonable request.

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
