# Peer review of "Ultrasonic Diagnosis of Intestinal Obstruction in Neonates-Original Article"

_diagnostics, 2023, doi:10.3390/diagnostics13050995_

Round 1

Reviewer 1 Report

The article shows the outcome of a thoroughly investigated long-year study. The work as it is presented has a high scientific soundness in the discussion. But other than that the remainig sections lack essential information. The structure and the presentation within the article needs to be improved. New findings of this study should be clearly pointed out and distinguishable for the readers.

1) Introduction: The state of the art needs to be clearly presented. Some aspects of this regard are to be found in the discussion, but it is not clear why the state of the art is not sufficient, why this work is necessary or helpful or what new research questions are to be answered. Statements from literature should be included about the general incidence of these diseases in order to support the results.

2) Methods: This section lacks relevant information to reproduce and comprehend the results, especially on the technical aspect and the design of the study. The authors make no statements on which ultrasound systems were used (vendor, specification, probe) and how. Was the data collected by experienced pediatricians in the field? How were the cases chosen/filtered? 86 cases do not seem to be many for a study conducted over 13 years. This leaves the notion that only obvious or very acute cases could be identified. This aspect and the affect to this study needs to be thoroughly discussed in order to judge the results.

3) Figure 3a is ethical questionable. I doubt that this picture is necessary on a scientific scale.

4) Discussion: This is the strongest section of this article. However, I would suggest to structure it into sections. I suggest to make the discussions in closer references to the results presented in Sec. 3. Furthermore, some of the so far missing informations are given here, but this supports the fact that this article needs to be better structured. Reasons why some features are not easily detected by ultrasound should be discussed (line 201 and 242).

5) The conclusion is far to general, the actual conclusion is given in the end of section 4, showing again that the structure of the article is missing stringency and needs to be improved.

Reviewer 2 Report

Ultrasound is becoming more available and cheaper than traditional x-rays.  Currently Xrays and fluoroscopic studies are the gold standard for working up congenital bowel lesions.  This study demonstrates reasonable ability of US to detect bowel obstruction in neonates and also may provide additional information regarding the etiology of the bowel obstruction. 

The paper never addresses how US compares to standard xray modalities and that is a big weakness of the paper. Using the operative findings as the gold standard is interesting but was the decision made to operate based on the US or standard xrays/fluoro studies.  

Additionally the authors focus too much on congenital anomalies rather than the ability of the US imaging to detect small bowel obstruction.  For example CDH demonstrating herniated bowel into the chest cavity or pyloric stenosis.  These are completely separate entities and have no place in this paper.  

Author Response

Response: Thank you very much for your valuable comments on our research, which will be very beneficial to us to improve our research and articles. As you said, X-ray is indeed a traditional and very important method for the diagnosis of intestinal obstruction. X-ray is indispensable for the diagnosis of intestinal obstruction, especially in adults. Contrast radiography and lower gastrointestinal radiography are also very useful for locating the obstruction site and determining the etiology of intestinal obstruction. In this study, we only use ultrasound to diagnose intestinal obstruction in newborns, because the abdominal wall of children, especially newborns, is very thin, and ultrasound can clearly observe the internal structure. , not only can diagnose neonatal intestinal obstruction very accurately, but most of them can accurately determine the cause of neonatal intestinal obstruction. In any case, our research did not mention that the comparison between ultrasound and X-ray is our incompleteness. A description of X-rays will be added in the revised manuscript. This suggestion pointed out the direction for our next research. In future research, we will use the operating results as the gold standard for judging whether the diagnosis of ultrasound and X-ray is correct, and compare the effectiveness of ultrasound and X-ray in diagnosing intestinal obstruction and its etiology sexual difference. Ultrasound for the diagnosis of intestinal obstruction shows that the proximal intestinal tube is dilated and the distal intestinal tube is shrunk. If there are such typical ultrasonic findings and corresponding clinical manifestations, ultrasound can easily make a diagnosis of intestinal obstruction. This manifestation is single and very difficult. Difficult to expand a lot of content. Another important content of this study is to use ultrasound to determine the location and cause of intestinal obstruction, because the research object is newborns, the most common cause of intestinal obstruction in newborns is congenital structural abnormalities, and such structural abnormalities are diverse. Many of them can be clearly observed by pediatric ultrasound, so the content of the discussion is too much. Thank you again for your careful guidance.

Changes: We have added the description of X-rays to the manuscript. (Line 191-202)

“X-ray is a traditional and essential method for the diagnosis of ileus, especially in adults. In addition to conventional standing plain abdominal radiography, upper and lower gastrointestinal radiography are also very useful for the location of ileus and the determination of its cause. Ultrasonography which is radiation-free is more suitable for children, especially newborns. The abdominal wall of newborns is so thin that sonographer can clearly observe the internal structure. After a systematic and in-depth study, an experienced sonographer can not only diagnose intestinal obstruction but also identify most of its causes. In our study, although the children also underwent X-ray examinations, we took the surgical results as the gold standard and only collected ultrasonic data to study the accuracy of ultrasonic diagnosis of neonatal ileus. The results show that ultrasound is a reliable alternative method for diagnosing gastrointestinal obstruction in neonates.”

Reviewer 3 Report

The authors performed a retrospective study of all neonatal intestinal obstructions between 2009 and 2022 and compared accuracy of ultrasound diagnosis of intestinal obstruction to surgical findings. The authors report an accuracy of 91% to diagnose obstruction and an accuracy of 84% to identify the correct etiology of this obstruction as compared to surgical findings.

1. Around 7 cases of neonatal obstruction per year appears low overall as the authors mention they included all neonatal cases during this period. There was only 1 case of pyloric stenosis in 13 years?

2. The authors state that only abdominal ultrasound was used for diagnosis before the patients were taken to the OR. No abdominal radiograph was obtained? It is hard to believe there was no work-up of the patients besides the abdominal ultrasound at all.

3. Please have an English speaker edit the manuscript and correct minor errors in the table and the paragraphs. Sentences and expressions, such as "Hypertension of the proximal bowel" and "Early surgery can often obtain good efficacy and prognosis", should be corrected. I do not completely understand the description of sonographic technical details in the discussion, likely due to poor language skills.

4. I do not understand that the authors differentiate between duodenal stricture, duodenal atresia and annular pancreas as causes of duodenal obstruction. Do they mean duodenal atresia without annular pancreas and duodenal web and duodenal atresia with annular pancreas? I am a bit baffled by the ability to differentiate those causes with the stated accuracy in the author's practice, which is certainly unheard of in the Western world.

5. I am taken back by the results of this study. I cannot even image that the cause of neonatal intestinal obstruction can be diagnosed that accurately with ultrasound, especially the level of atresia or the cause of duodenal obstruction. Other reported findings such as volvulus may be more straight forward. I would like much better technical details of the ultrasound studies reported in the method section (what ultrasound machines, probes, technique of their scans, are the ultrasounds performed by radiologists or techs or who else?). The discussion must be much more in detail about ultrasound findings in the different causes and its differential diagnoses as possible seen on ultrasound.

6. The authors have not convinced my about the results of their study. They need to work on this study to make it valuable or, possibly, reproducible for other interested readers who want to train themselves to copy those results.

Round 2

Reviewer 1 Report

Thank you for your revision and thorough responses. The paper improved a lot by providing more details. As you clearly stated, you want to help other pediatricians in ultrasound based diagnostics. Therefore, detailed methods and critical discussions are very important for the re-use of your results. Just a few last and minor remarks:

Sec. 1: comparative studies are still not mentioned / discussed to embed this work into the state of the art.

Sec. 2, new paragraph: Quite informative and good for the re-use of other medical doctors! However, the procedure could be presented better to improve readability. Consider language editing, a flow chart or a visual presentation.

- Sec. 2, line 65: 3-11 MHz sounds like a range of transducers were used, as that frequency range can hardly be provided by a single probe. Which types were used? Were the probes switched depending on the imaged feature? Please add more details for a thorough reproduction.

- Sec. 5: you can still strenghten the paper's impact by drawing more thorough conclusions from your discussion section.

Author Response

Thank you very much for your recognition of our work. We have revised and improved the paper according to your suggestion. 

Please see the attachment for details. The  revised portions are marked in highlights with green in the paper. 

Reviewer 3 Report

Thank you for updating your manuscript. It adds clarity to the presentation and your considerable expertise.

Author Response

Thank you for your recognition of our work. The comments provided were extremely valuable in our effort to revise and improve our paper and gave our authors vital guidance to improve our research.